# ISOMETRIC AUTOENCODERS

## ABSTRACT

High dimensional data is often assumed to be concentrated on or near a low-dimensional manifold. Autoencoders (AE) is a popular technique to learn representations of such data by pushing it through a neural network with a low dimension bottleneck while minimizing a reconstruction error. Using high capacity AE often leads to a large collection of minimizers, many of which represent a low dimensional manifold that fits the data well but generalizes poorly.

Two sources of bad generalization are: extrinsic, where the learned manifold possesses extraneous parts that are far from the data; and intrinsic, where the encoder and decoder introduce arbitrary distortion in the low dimensional parameterization. An approach taken to alleviate these issues is to add a regularizer that favors a particular solution; common regularizers promote sparsity, small derivatives, or robustness to noise.

In this paper, we advocate an isometry (i.e., local distance preserving) regularizer. Specifically, our regularizer encourages: (i) the decoder to be an isometry; and (ii) the encoder to be the decoder's pseudo-inverse, that is, the encoder extends the inverse of the decoder to the ambient space by orthogonal projection. In a nutshell, (i) and (ii) fix both intrinsic and extrinsic degrees of freedom and provide a non-linear generalization to principal component analysis (PCA).

Experimenting with the isometry regularizer on dimensionality reduction tasks produces useful low-dimensional data representations.

## 1 INTRODUCTION

A common assumption is that high dimensional data $\mathcal{X} \subset \mathbb{R}^D$ is sampled from some distribution $p$ concentrated on, or near, some lower $d$-dimensional submanifold $\mathcal{M} \subset \mathbb{R}^D$, where $d < D$. The task of estimating $p$ can therefore be decomposed into: (i) approximate the manifold $\mathcal{M}$; and (ii) approximate $p$ restricted to, or concentrated near $\mathcal{M}$.

In this paper we focus on task (i), mostly known as *manifold learning*. A common approach to approximate the $d$-dimensional manifold $\mathcal{M}$, e.g., in (Tenenbaum et al., 2000; Roweis & Saul, 2000; Belkin & Niyogi, 2002; Maaten & Hinton, 2008; McQueen et al., 2016; McInnes et al., 2018), is to embed $\mathcal{X}$ in $\mathbb{R}^d$. This is often done by first constructing a graph $\mathcal{G}$ where nearby samples in $\mathcal{X}$ are conngected by edges, and second, optimizing for the locations of the samples in $\mathbb{R}^d$ striving to minimize edge length distortions in $\mathcal{G}$.

Autoencoders (AE) can also be seen as a method to learn low dimensional manifold representation of high dimensional data $\mathcal{X}$. AE are designed to reconstruct $\mathcal{X}$ as the image of its low dimensional embedding. When restricting AE to linear encoders and decoders it learns linear subspaces; with mean squared reconstruction loss they reproduce principle component analysis (PCA). Using higher capacity neural networks as the encoder and decoder, allows complex manifolds to be approximated. To avoid overfitting, different regularizers are added to the AE loss. Popular regularizers include sparsity promoting (Ranzato et al., 2007; 2008; Glorot et al., 2011), contractive or penalizing large derivatives (Rifai et al., 2011a;b), and denoising (Vincent et al., 2010; Poole et al., 2014). Recent AE regularizers directly promote distance preservation of the encoder (Pai et al., 2019; Peterfreund et al., 2020).

In this paper we advocate a novel AE regularization promoting isometry (i.e., local distance preservation), called Isometric-AE (I-AE). Our key idea is to promote the decoder to be isometric, and the encoder to be its *pseudo-inverse*. Given an isometric decoder $\mathbb{R}^d \to \mathbb{R}^D$, there is no well-defined

inverse $\mathbb{R}^D \to \mathbb{R}^d$; we define the pseudo-inverse to be a projection on the image of the decoder composed with the inverse of the decoder restricted to its image.

Locally, the I-AE regularization therefore encourages: (i) the differential of the decoder $\boldsymbol{A} \in \mathbb{R}^{D \times d}$ to be an isometry, i.e., $\boldsymbol{A}^T \boldsymbol{A} = \boldsymbol{I}_d$, where $\boldsymbol{I}_d$ is the $d \times d$ identity matrix; and (ii) the differential of the encoder, $\boldsymbol{B} \in \mathbb{R}^{d \times D}$ to be the pseudo-inverse (now in the standard linear algebra sense) of the differential of the decoder $\boldsymbol{A} \in \mathbb{R}^{D \times d}$, namely, $\boldsymbol{B} = \boldsymbol{A}^+$. In view of (i) this implies $\boldsymbol{B} = \boldsymbol{A}^T$. This means that *locally* our decoder and encoder behave like PCA, where the encoder and decoder are linear transformations satisfying (i) and (ii); That is, the PCA encoder can be seen as a composition of an orthogonal projection on the linear subspace spanned by the decoder, followed by an orthogonal transformation (isometry) to the low dimensional space.

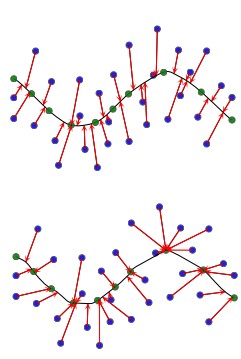

Figure 1: Top: I-AE; bottom: CAE.

In a sense, our method can be seen as a version of denoising/contractive AEs (DAE/CAE, respectively). DAE and CAE promote a projection from the ambient space onto the data manifold, but can distort distances and be non-injective. Locally, using differentials again, projection on the learned manifold means $(\boldsymbol{A}\boldsymbol{B})^2 = \boldsymbol{A}\boldsymbol{B}$. Indeed, as can be readily checked conditions (i) and (ii) above imply $\boldsymbol{A}(\boldsymbol{B}\boldsymbol{A})\boldsymbol{B} = \boldsymbol{A}\boldsymbol{B}$. This means that I-AE also belongs to the same class of DAE/CAE, capturing the variations in tangent directions of the data, $\mathcal{M}$, while ignoring orthogonal variations which often represent noise (Vincent et al., 2010; Alain & Bengio, 2014). The benefit in I-AE is that its projection on the data manifold is locally an isometry, preserving distances and sampling the learned manifold evenly. That is, I-AE does not shrink or expand the space; locally, it can be imagined as an orthogonal linear transformation. The inset shows results of a simple experiment comparing contractive AE (CAE-bottom) and isometric AE (I-AE-top). Both AEs are trained on the green data points; the red arrows depict projection of points (in blue) in vicinity of the data onto the learned manifold (in black) as calculated by applying the encoder followed by the decoder. Note that CAE indeed projects on the learned manifold but not evenly, tending to shrink space around data points; in contrast I-AE provides a more even sampling of the learned manifold.

Experiments confirm that optimizing the I-AE loss results in a close-to-isometric encoder/decoder explaining the data. We further demonstrate the efficacy of I-AE for dimensionality reduction of different standard datatsets, showing its benefits over manifold learning and other AE baselines.

## 2 RELATED WORKS

**Manifold learning.** Manifold learning generalizes classic dimensionality reduction methods such as PCA (F.R.S., 1901) and MDS (Kruskal, 1964; Sammon, 1969), by aiming to preserve the local geometry of the data. Tenenbaum et al. (2000) use the nn-graph to approximate the geodesic distances over the manifold, followed by MDS to preserve it in the lower dimension. Roweis & Saul (2000); Belkin & Niyogi (2002); Donoho & Grimes (2003) use spectral methods to minimize different distortion energy functions over the graph matrix. Coifman et al. (2005); Coifman & Lafon (2006) approximate the heat diffusion over the manifold by a random walk over the nn-graph, to gain a robust distance measure on the manifold. Stochastic neighboring embedding algorithms (Hinton & Roweis, 2003; Maaten & Hinton, 2008) captures the local geometry of the data as a mixture of Gaussians around each data points, and try to find a low dimension mixture model by minimizing the KL-divergence. In a relatively recent work, McInnes et al. (2018) use iterative spectral and embedding optimization using fuzzy sets. Several works tried to adapt classic manifold learning ideas to neural networks and autoencoders. Pai et al. (2019) suggest to embed high dimensional points into a low dimension with a neural network by constructing a metric between pairs of data points and minimizing the metric distortion energy. Kato et al. (2019) suggest to learn an isometric decoder by using noisy latent variables. They prove under certain conditions that it encourages isometric decoder. Peterfreund et al. (2020) suggest autoencoders that promote the isometry of the encoder over the data by approximating its differential gram matrix using sample covariance matrix. Zhan et al. (2018) encourage distance preserving autoencoders by minimizing metric distortion energy in common feature space.

**Modern autoencoders.** There is an extensive literature on extending autoencoders to a generative model (task (ii) in section 1). That is, learning a probability distribution in addition to approximating the data manifold $\mathcal{M}$. Variational autoencoder (VAE) Kingma & Welling (2014) and its variants Makhzani et al. (2015); Burda et al. (2016); Sønderby et al. (2016); Higgins et al. (2017); Tolstikhin et al. (2018); Park et al. (2019); Zhao et al. (2019) are examples to such methods. In essence, these methods augment the AE structure with a learned probabilistic model in the low dimensional (latent) space $\mathbb{R}^d$ that is used to approximate the probability $P$ that generated the observed data $\mathcal{X}$. More relevant to our work, are recent works suggesting regularizers for deterministic autoencoders that together with ex-post density estimation in latent space forms a generative model. Ghosh et al. (2020) suggested to reduce the decoder degrees of freedom, either by regularizing the norm of the decoder weights or the norm of the decoder differential. Other regularizers of the differential of the decoder, aiming towards a deterministic variant of VAE, were recently suggested in Kumar & Poole (2020); Kumar et al. (2020). In contrast to our method, these methods do not regularize the encoder explicitly.

## 3 ISOMETRIC AUTOENCODERS

We consider high dimensional data points $\mathcal{X} = \{\boldsymbol{x}_i\}_{i=1}^n \subset \mathbb{R}^D$ sampled from some probability distribution $P(\boldsymbol{x})$ in $\mathbb{R}^D$ concentrated on or near some $d$ dimensional submanifold $\mathcal{M} \subset \mathbb{R}^D$, where $d < D$.

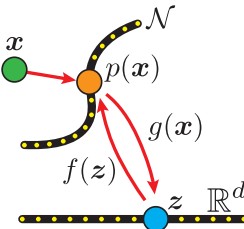

Our goal is to compute *isometric autoencoder* (I-AE) defined as follows. Let $g : \mathbb{R}^D \to \mathbb{R}^d$ denote the encoder, and $f : \mathbb{R}^d \to \mathbb{R}^D$ the decoder; $\mathcal{N}$ is the learned manifold, i.e., the image of the decoder, $\mathcal{N} = f(\mathbb{R}^d)$. I-AE is defined by the following requirements:

(i) The data $\mathcal{X}$ is close to $\mathcal{N}$.

(ii) $f$ is an isometry.

(iii) $g$ is the pseudo-inverse of $f$.

Figure 2: I-AE.

Figure 2 is an illustration of I-AE. Let $\theta$ denote the parameters of $f$, and $\phi$ the parameters of $g$. We enforce the requirements (i)-(iii) by prescribing a loss function $L(\theta, \phi)$ and optimize it using standard stochastic gradient descent (SGD). We next break down the loss $L$ to its different components.

Condition (i) is promoted with the standard reconstruction loss in AE:

$$L_{\text{rec}}(\theta, \phi) = \frac{1}{n} \sum_{i=1}^n \|f(g(\boldsymbol{x}_i)) - \boldsymbol{x}_i\|^2, \tag{1}$$

where $\|\cdot\|$ is the 2-norm.

Before handling conditions (ii),(iii) let us first define the notions of isometry and pseudo-inverse. A differentiable mapping $f$ between the euclidean spaces $\mathbb{R}^d$ and $\mathbb{R}^D$ is a local isometry if it has an orthogonal differential matrix $df(\boldsymbol{z}) \in \mathbb{R}^{D \times d}$,

$$df(\boldsymbol{z})^T df(\boldsymbol{z}) = \boldsymbol{I}_d, \tag{2}$$

where $\boldsymbol{I}_d \in \mathbb{R}^{d \times d}$ is the identity matrix, and $df(\boldsymbol{z})_{ij} = \frac{\partial f^i}{\partial z_j}(\boldsymbol{z})$. A local isometry which is also a diffeomorphism is a global isometry. Restricting the decoder to isometry is beneficial for several reasons. First, Nash-Kuiper Embedding Theorem Nash (1956) asserts that non-expansive maps can be approximated arbitrary well with isometries if $D \geq d + 1$ and hence promoting an isometry does not limit the expressive power of the decoder. Second, the low dimensional representation of the data computed with an isometric encoder preserves the geometric structure of the data. In particular volume, length, angles and probability densities are preserved between the low dimensional representation $\mathbb{R}^d$, and the learned manifold $\mathcal{N}$. Lastly, for a fixed manifold $\mathcal{N}$ there is a huge space of possible decoders such that $\mathcal{N} = f(\mathbb{R}^d)$. For isometric $f$, this space is reduced considerably: Indeed, consider two isometries parameterizing $\mathcal{N}$, i.e., $f_1, f_2 : \mathbb{R}^d \to \mathcal{N}$. Then, since composition of isometries is an isometry we have that $f_2^{-1} \circ f_1 : \mathbb{R}^d \to \mathbb{R}^d$ is a dimension-preserving isometry and hence a rigid motion. That is, all decoders of the same manifold are the same up to a rigid motion.

For the encoder the situation is different. Since $D > d$ the encoder $g$ cannot be an isometry in the standard sense. Therefore we ask $g$ to be the *pseudo-inverse* of $f$. For that end we define the projection operator $\boldsymbol{p}$ on a submanifold $\mathcal{N} \subset \mathbb{R}^D$ as

$$\boldsymbol{p}(\boldsymbol{x}) = \arg \min_{\boldsymbol{x}' \in \mathcal{N}} \|\boldsymbol{x} - \boldsymbol{x}'\| .$$

Note that the closest point is not generally unique, however the Tubular Neighborhood Theorem (see e.g., Theorem 6.24 in Lee (2013)) implies uniqueness for points $\boldsymbol{x}$ sufficiently close to the manifold $\mathcal{N}$.

**Definition 1.** *We say the $g$ is the* pseudo-inverse *of $f$ if $g$ can be written as $g = f^{-1} \circ \boldsymbol{p}$, where $\boldsymbol{p}$ is the projection on $\mathcal{N} = f(\mathbb{R}^d)$.*

Consequently, if $g$ is the pseudo-inverse of an isometry $f$ then it extends the standard notion of isometry by projecting every point on a submanifold $\mathcal{N}$ and then applying an isometry between the $d$-dimensional manifolds $\mathcal{N}$ and $\mathbb{R}^d$. See Figure 2 for an illustration.

**First-order characterization.** To encourage $f, g$ to satisfy the (local) isometry and the pseudo-inverse properties (resp.) we will first provide a first-order (necessary) characterization using their differentials:

**Theorem 1.** *Let $f$ be a decoder and $g$ an encoder satisfying conditions (ii),(iii). Then their differentials $\boldsymbol{A} = df(\boldsymbol{z}) \in \mathbb{R}^{D \times d}$, $\boldsymbol{B} = dg(f(\boldsymbol{z})) \in \mathbb{R}^{d \times D}$ satisfy*

$$\boldsymbol{A}^T \boldsymbol{A} = \boldsymbol{I}_d \tag{3}$$

$$\boldsymbol{B}\boldsymbol{B}^T = \boldsymbol{I}_d \tag{4}$$

$$\boldsymbol{B} = \boldsymbol{A}^T \tag{5}$$

The theorem asserts that the differentials of the encoder and decoder are orthogonal (rectangular) matrices, and that the encoder is the pseudo-inverse of the differential of the decoder. Before proving this theorem, let us first use it to construct the relevant losses for promoting the isometry of $f$ and pseudo-inverse $g$. We need to promote conditions (3), (4), (5). Since we want to avoid computing the full differentials $\boldsymbol{A} = df(\boldsymbol{z})$, $\boldsymbol{B} = dg(f(\boldsymbol{z}))$, we will replace (3) and (4) with stochastic estimations based on the following lemma: denote the unit $d - 1$-sphere by $\mathcal{S}^{d-1} = \{\boldsymbol{z} \in \mathbb{R}^d | \, \|\boldsymbol{z}\| = 1\}$.

**Lemma 1.** *Let $\boldsymbol{A} \in \mathbb{R}^{D \times d}$, where $d \leq D$. If $\|\boldsymbol{A}\boldsymbol{u}\| = 1$ for all $\boldsymbol{u} \in \mathcal{S}^{d-1}$, then $\boldsymbol{A}$ is column-orthogonal, that is $\boldsymbol{A}^T \boldsymbol{A} = \boldsymbol{I}_d$.*

Therefore, the isometry promoting loss, encouraging (3), is defined by

$$L_{\text{iso}}(\theta) = \mathbb{E}_{\boldsymbol{z},\boldsymbol{u}} \Big( \|df(\boldsymbol{z})\boldsymbol{u}\| - 1 \Big)^2 , \tag{6}$$

where $\boldsymbol{z} \sim P_{\text{iso}}(\mathbb{R}^d)$, and $P_{\text{iso}}(\mathbb{R}^d)$ is a probability measure on $\mathbb{R}^d$; $\boldsymbol{u} \sim P(\mathcal{S}^{d-1})$, and $P(\mathcal{S}^{d-1})$ is the standard rotation invariant probability measure on the $d - 1$-sphere $\mathcal{S}^{d-1}$. The pseudo-inverse promoting loss, encouraging (4) would be

$$L_{\text{piso}}(\phi) = \mathbb{E}_{\boldsymbol{x},\boldsymbol{u}} \Big( \|\boldsymbol{u}^T dg(\boldsymbol{x})\| - 1 \Big)^2 , \tag{7}$$

where $\boldsymbol{x} \sim P(\mathcal{M})$ and $\boldsymbol{u} \sim P(\mathcal{S}^{d-1})$. As usual, the expectation with respect to $P(\mathcal{M})$ is computed empirically using the data samples $\mathcal{X}$.

Lastly, (5) might seem challenging to enforce with neural networks, however the orthogonality of $\boldsymbol{A}, \boldsymbol{B}$ can be leveraged to replace this loss with a more tractable loss asking the encoder is merely the inverse of the decoder over its image:

**Lemma 2.** *Let $\boldsymbol{A} \in \mathbb{R}^{D \times d}$, and $\boldsymbol{B} \in \mathbb{R}^{d \times D}$. If $\boldsymbol{A}^T \boldsymbol{A} = \boldsymbol{I}_d = \boldsymbol{B}\boldsymbol{B}^T$ and $\boldsymbol{B}\boldsymbol{A} = \boldsymbol{I}_d$ then $\boldsymbol{B} = \boldsymbol{A}^+ = \boldsymbol{A}^T$.*

Fortunately, this is already taken care of by the reconstruction loss: since low reconstruction loss in equation 1 forces the encoder and the decoder to be the inverse of one another over the data manifold, i.e., $g(f(\boldsymbol{z})) = \boldsymbol{z}$, it encourages $\boldsymbol{B}\boldsymbol{A} = \boldsymbol{I}_d$ and therefore, by Lemma 2, automatically encourages equation 5. Note that invertability also implies bijectivity of the encoder/decoder restricted to the data manifold, pushing for global isometries (rather than local). Summing all up, we define our loss for I-AE by

$$L(\theta, \phi) = L_{\text{rec}}(\theta, \phi) + \lambda_{\text{iso}} \left( L_{\text{iso}}(\theta) + L_{\text{piso}}(\phi) \right) , \tag{8}$$

where $\lambda_{\text{iso}}$ is a parameter controlling the isometry-reconstruction trade-off.

## 3.1 DETAILS AND PROOFS.

Let us prove Theorem 1 characterizing the relation of the differentials of isometries and pseudo-isometries, $\boldsymbol{A} = df(\boldsymbol{z}) \in \mathbb{R}^{D \times d}$, $\boldsymbol{B} = dg(f(\boldsymbol{z})) \in \mathbb{R}^{d \times D}$. First, by definition of isometry (equation 2), $\boldsymbol{A}^T \boldsymbol{A} = \boldsymbol{I}_d$. We denote by $T_{\boldsymbol{x}} \mathcal{N}$ the $d$-dimensional tangent space to $\mathcal{N}$ at $\boldsymbol{x} \in \mathcal{N}$; accordingly, $T_{\boldsymbol{x}} \mathcal{N}^{\perp}$ denotes the normal tangent space.

**Lemma 3.** *The differential $d\boldsymbol{p}(\boldsymbol{x}) \in \mathbb{R}^{D \times D}$ at $\boldsymbol{x} \in \mathcal{N}$ of the projection operator $\boldsymbol{p} : \mathbb{R}^D \to \mathcal{N}$ is*

$$d\boldsymbol{p}(\boldsymbol{x})\boldsymbol{u} = \begin{cases} \boldsymbol{u} & \boldsymbol{u} \in T_{\boldsymbol{x}} \mathcal{N} \\ 0 & \boldsymbol{u} \in T_{\boldsymbol{x}} \mathcal{N}^{\perp} \end{cases} \tag{9}$$

*That is, $d\boldsymbol{p}(\boldsymbol{x})$ is the orthogonal projection on the tangent space of $\mathcal{N}$ at $\boldsymbol{x}$.*

*Proof.* First, consider the squared distance function to $\mathcal{N}$ defined by $\eta(\boldsymbol{x}) = \frac{1}{2} \min_{\boldsymbol{x}' \in \mathcal{N}} \|\boldsymbol{x} - \boldsymbol{x}'\|^2$. The envelope theorem implies that $\nabla \eta(\boldsymbol{x}) = \boldsymbol{x} - \boldsymbol{p}(\boldsymbol{x})$. Differentiating both sides and rearranging we get $d\boldsymbol{p}(\boldsymbol{x}) = \boldsymbol{I}_D - \nabla^2 \eta(\boldsymbol{x})$. As proved in Ambrosio & Soner (1994) (Theorem 3.1), $\nabla^2 \eta(\boldsymbol{x})$ is the orthogonal projection on $T_{\boldsymbol{x}} \mathcal{N}^{\perp}$. $\square$

Let $\boldsymbol{x} = f(\boldsymbol{z}) \in \mathcal{N}$. Since $\boldsymbol{x} \in \mathcal{N}$ we have $\boldsymbol{p}(\boldsymbol{x}) = \boldsymbol{x}$. Condition (iii) asserts that $g(\boldsymbol{y}) = f^{-1}(\boldsymbol{p}(\boldsymbol{y}))$; taking the derivative at $\boldsymbol{y} = \boldsymbol{x}$ we get $dg(\boldsymbol{x}) = df^{-1}(\boldsymbol{x})d\boldsymbol{p}(\boldsymbol{x})$. Lemma 3 implies that $d\boldsymbol{p}(\boldsymbol{x}) = \boldsymbol{A}\boldsymbol{A}^T$, since $\boldsymbol{A}\boldsymbol{A}^T$ is the orthogonal projection on $\mathcal{T}_{\boldsymbol{x}} \mathcal{N}$. Furthermore, $df^{-1}(\boldsymbol{x})$ restricted to $\text{Im}(\boldsymbol{A})$ is $\boldsymbol{A}^T$. Putting this together we get $\boldsymbol{B} = dg(\boldsymbol{x}) = \boldsymbol{A}^T \boldsymbol{A}\boldsymbol{A}^T = \boldsymbol{A}^T$. This implies that $\boldsymbol{B}\boldsymbol{B}^T = \boldsymbol{I}_d$, and that $\boldsymbol{B} = \boldsymbol{A}^+ = \boldsymbol{A}^T$. This concludes the proof of Theorem 1. $\square$

*Proof of Lemma 1.* Writing the SVD of $\boldsymbol{A} = \boldsymbol{U}\Sigma\boldsymbol{V}^T$, where $\Sigma = \text{diag}(\sigma_1, \dots, \sigma_d)$ are the singular values of $\boldsymbol{A}$, we get that $\sum_{i=1}^d \sigma_i^2 v_i^2 = 1$ for all $\boldsymbol{v} \in \mathcal{S}^{d-1}$. Plugging $\boldsymbol{v} = \boldsymbol{e}_j$, $j \in [d]$ (the standard basis) we get that all $\sigma_i = 1$ for $i \in [d]$ and $\boldsymbol{A} = \boldsymbol{U}\boldsymbol{V}^T$ is orthogonal as claimed. $\square$

*Proof of Lemma 2.* Let $\boldsymbol{U} = [\boldsymbol{A}, \boldsymbol{V}]$, $\boldsymbol{V} \in \mathbb{R}^{D \times (D-d)}$, be a completion of $\boldsymbol{A}$ to an orthogonal matrix in $\mathbb{R}^{D \times D}$. Now, $\boldsymbol{I}_d = \boldsymbol{B}\boldsymbol{U}\boldsymbol{U}^T\boldsymbol{B}^T = \boldsymbol{I}_d + \boldsymbol{B}\boldsymbol{V}\boldsymbol{V}^T\boldsymbol{B}^T$, and since $\boldsymbol{B}\boldsymbol{V}\boldsymbol{V}^T\boldsymbol{B}^T \succeq 0$ this means that $\boldsymbol{B}\boldsymbol{V} = 0$, that is $\boldsymbol{B}$ takes to null the orthogonal space to the column space of $\boldsymbol{A}$. A direct computation shows that $\boldsymbol{B}\boldsymbol{U} = \boldsymbol{A}^T\boldsymbol{U}$ which in turn implies $\boldsymbol{B} = \boldsymbol{A}^T = \boldsymbol{A}^+$. $\square$

**Implementation.** Implementing the losses in equation 6 and equation 7 requires making a choice for the probability densities and approximating the expectations. We take $P_{\text{iso}}(\mathbb{R}^d)$ to be either uniform or gaussian fit to the latent codes $g(\mathcal{X})$; and $P(\mathcal{M})$ is approximated as the uniform distribution on $\mathcal{X}$, as mentioned above. The expectations are estimated using Monte-Carlo sampling. That is, at each iteration we draw samples $\hat{\boldsymbol{x}} \in \mathcal{X}$, $\hat{\boldsymbol{z}} \sim P_{\text{iso}}(\mathbb{R}^d)$, $\hat{\boldsymbol{u}} \sim P(\mathcal{S}^{d-1})$ and use the approximations

$$L_{\text{iso}}(\theta) \approx \left( \|df(\hat{\boldsymbol{z}})\hat{\boldsymbol{u}}\| - 1 \right)^2$$

$$L_{\text{piso}}(\phi) \approx \left( \|\hat{\boldsymbol{u}}^T dg(\hat{\boldsymbol{x}})\| - 1 \right)^2$$

The right differential multiplication $df(\hat{\boldsymbol{z}})\hat{\boldsymbol{u}}$ and left differential multiplication $\hat{\boldsymbol{u}}^T dg(\hat{\boldsymbol{x}})$ are computed using forward and backward mode automatic differentiation (resp.). Their derivatives with respect to the networks' parameters $\theta, \phi$ are computed by another backward mode automatic differentiation.

## 4 EXPERIMENTS

### 4.1 EVALUATION

We start by evaluating the effectiveness of our suggested I-AE regularizer, addressing the following questions: (i) does our suggested loss $L(\theta, \phi)$ in equation 8 drive I-AE training to converge to an isometry? (ii) What is the effect of the $L_{\text{piso}}$ term? In particular, does it encourage better manifold approximations as conjectured? To that end, we examined the I-AE training on data points $\mathcal{X}$ sampled uniformly from 3D surfaces with known global parameterizations. Figure 3 shows qualitative comparison of the learned embeddings for various AE regularization techniques: Vanilla autoencoder (AE); Contractive autoencoder (CAE) (Rifai et al., 2011b); Contractive autoencoder with decoder

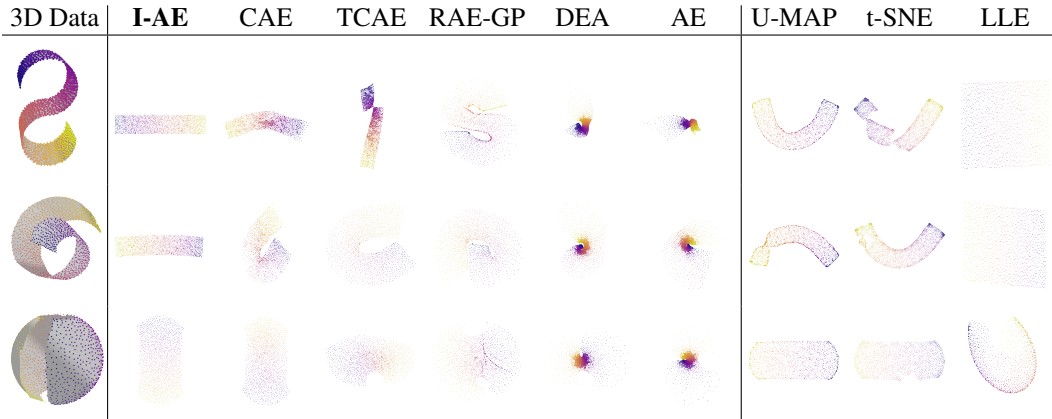

| 3D Data | **I-AE** | CAE | TCAE | RAE-GP | DEA | AE | U-MAP | t-SNE | LLE |
|---------|----------|-----|------|--------|-----|-----|-------|-------|-----|

Figure 3: Evaluation of $3D \rightarrow 2D$ embeddings.

weights tied to the encoder weights (TCAE) (Rifai et al., 2011a); Gradient penalty on the decoder (RAE-GP) (Ghosh et al., 2020); and Denoising autoencoder with gaussian noise (DAE) (Vincent et al., 2010). For fairness in evaluation, all methods were trained using the same training hyper-parameters. See Appendix for the complete experiment details including mathematical formulation of the different AE regularizers. In addition, we compared against popular classic manifold learning techniques: U-MAP (McInnes et al., 2018), t-SNE (Maaten & Hinton, 2008) and LLE. (Roweis & Saul, 2000). The results demonstrate that I-AE is able to learn an isometric embedding, showing some of the advantages in our method: sampling density and distances between input points is preserved in the learned low dimensional space.

In addition, for the AE methods, we quantitatively evaluate how close is the learnt decoder to an isometry. For this purpose, we triangulate a grid of planar points $\{z_i\} \subset \mathbb{R}^2$. We denote by $\{e_{ij}\}$ the triangles edges incident to grid points $z_i$ and $z_j$. Then, we measured the edge

|  | I-AE | CAE | TCAE | RAE-GP | DAE | AE |
|--|------|-----|------|--------|-----|-----|
| S Shape | **0.03** | 0.36 | 0.26 | 1.22 | 2.53 | 1.85 |
| Swiss Roll | **0.02** | 1.00 | 0.38 | 1.75 | 1.80 | 1.63 |
| Open Sphere | **0.07** | 0.21 | 0.21 | 0.50 | 1.09 | 1.29 |

Table 1: Std of $\{l_{ij}\}$.

lengths ratio, $l_{ij} = \|f(z_i) - f(z_j)\|/\|e_{ij}\|$ expected to be $\approx 1$ for all edges $e_{ij}$ in an isometry. In Table 1 we log the standard deviation (Std) of $\{l_{ij}\}$ for I-AE compared to other regularized AEs. For a fair comparison, we scaled $z_i$ so the mean of $l_{ij}$ is 1 in all experiments. As can be seen in the table, the distribution of $\{l_{ij}\}$ for I-AE is significantly more concentrated than the different AE baselines.

Finally, although $L_{\text{iso}}$ is already responsible for learning an isometric decoder, the pseudo-inverse encoder (enforced by the loss $L_{\text{piso}}$) helps it converge to simpler solutions. We ran AE training with and without the $L_{\text{piso}}$ term. Figure 4 shows in gray the learnt decoder surface, $\mathcal{N}$, without $L_{\text{piso}}$ (left), containing extra (unnatural) surface parts compared to the learnt surface with $L_{\text{piso}}$ (right). In both cases we expect (and achieve) a decoder approximating an isometry that passes through the input data points. Nevertheless, the

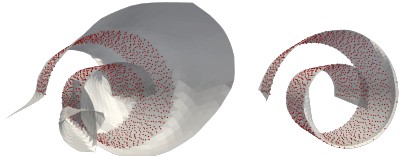

Figure 4: Decoder surfaces without $L_{\text{piso}}$ (left) and with (right).

pseudo-inverse loss restricts some of the degrees of freedom of the encoder which in turn leads to a simpler solution.

## 4.2 DATA VISUALIZATION

In this experiment we evaluate our method in the task of high dimension data visualization, i.e., reducing high dimensional data into two dimensional space. Usually the data is not assumed to lie on a manifold with such a low dimension, and it is therefore impossible to preserve all of its geometric properties. A common artifact when squeezing higher dimensional data into the plane is crowding (Maaten & Hinton, 2008), that is planar embedded points are crowded around the origin.

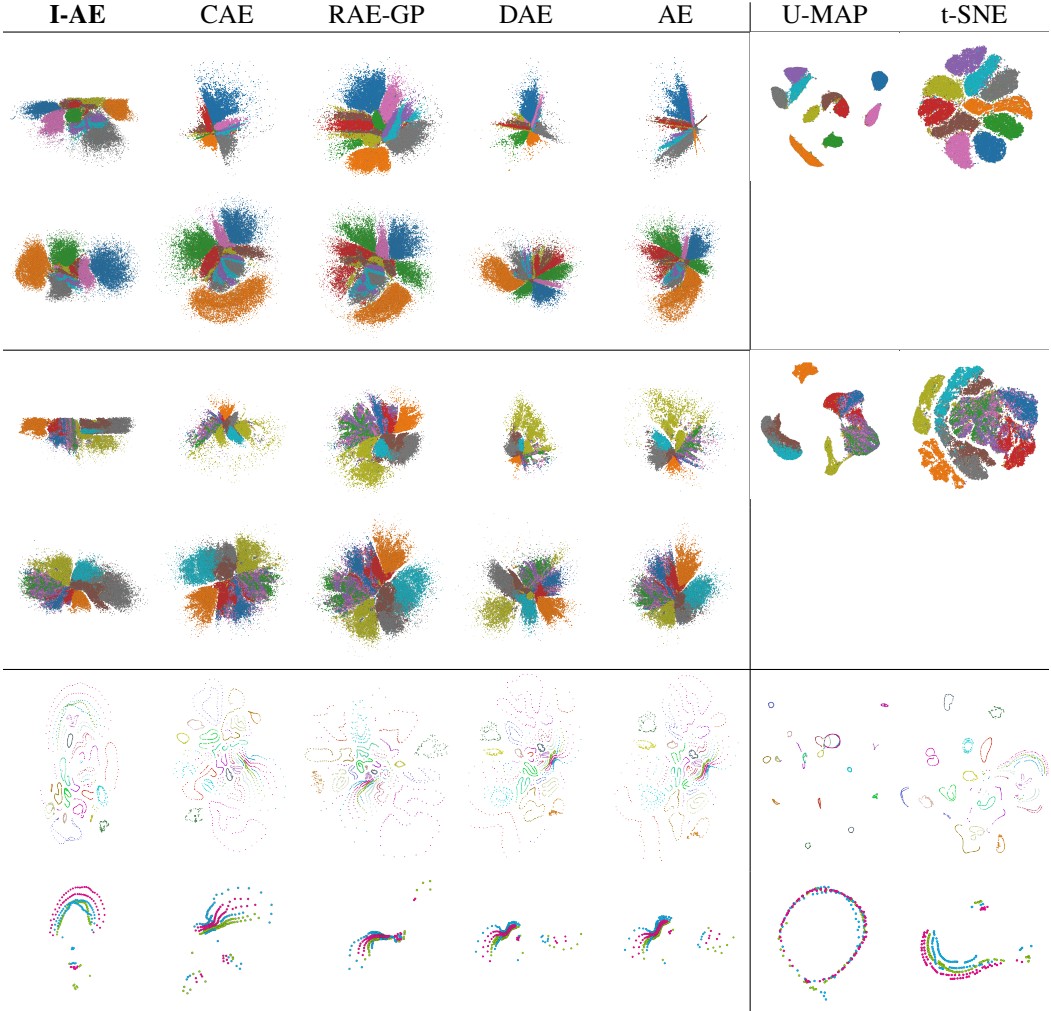

Figure 5: Results of data visualization experiment. Different colors indicate different ground turth labels/classes. Top shows MNIST: FC architecture of the encoder/decoder (top row), and CNN (bottom row); Middle shows FMNIST: FC (top row), and CNN (bottom row); Bottom shows COIL20 with CNN architecture, where zoom-ins of 3 classes are shown in the bottom row.

We evaluate our method on three standard datasets of images: MNIST (LeCun, 1998) (60k handwritten digits), Fashion-MNIST (60k Zalando's article images) (Xiao et al., 2017) and COIL20 (Nene et al., 1996) (20 different images of object rotated with 72 even rotations). For baselines we take: Vanilla AE; CAE; GP-RAE; DAE; U-MAP and t-SNE.

We use the same architecture for all auto-encoder methods on each dataset. MNIST and FMNIST we evaluated in two scenarios: (i) Both encoder and decoder are fully-connected (MLP) networks; and (ii) Both encoder and decoder are Convolutional Neural Network (CNN). For COIL20 dataset both encoder and decoder are Convolutional Neural Network. Full implementation details and hyper-parameters values can be found in the Appendix.

The results are presented in figure 5; where each embedded point $z$ is colored by its ground-truth class/label. We make several observation. First, in all the datasets our method is more resilient to crowding compared to the baseline AEs, and provide a more even spread. U-MAP and t-SNE produce better separated clusters. However, this separation can come at a cost: See the COIL20 result (third row) and blow-ups of three of the classes (bottom row). In this dataset we expect evenly spaced points that correspond to the even rotations of the objects in the images. Note (in the blow-ups) that U-MAP maps the three classes on top of each other (non-injectivity of the "encoder"), t-SNE is

somewhat better but does not preserve well the distance between pairs of data points (we expect them to be more or less equidistant in this dataset). In I-AE the rings are better separated and points are more equidistant; the baseline AEs tend to densify the points near the origin. Lastly, considering the inter and intra-class variations for the MNIST and FMNIST datasets, we are not sure that isometric embeddings are expected to produce strongly separated clusters as in U-MAP and t-SNE (e.g., think about similar digits of different classes and dissimilar digits of the same class with distances measured in euclidean norm).

### 4.3 DOWNSTREAM CLASSIFICATION

To quantitatively evaluate the unsupervised low-dimensional embedding computed with the I-AE we performed the following experiment: We trained simple classifiers on the embedded vectors computed by I-AE and baseline AEs and compared their performance (i.e., accuracy). Note that the process of learning the embedding is unsupervised and completely oblivious to the labels, which are used solely for training and testing the classifiers.

We evaluate on the same datasets as in Section 4.2: In MNIST and FMNIST we use the standard train-test split, and on COIL20 we split 75%-25% randomly. As AE baselines we take vanilla AE, CAE, DAE and RAE-GP, as described above. We repeat each experiment with 3 different latent dimensions, $\{16, 64, 256\}$, and use two different simple classification algorithms: linear *Support vector machines (SVM)* (Cortes & Vapnik, 1995) and $K$-*nearest neighbors (K-NN)*, with $K = 5$.

Table 2 logs the results, where for both types of classifiers I-AE outperforms the baseline AEs in almost all combinations, where the SVM experiments demonstrate larger margins in favor of I-AE. The results of the $K$-NN indicate that euclidean metric captures similarity in our embedding, and the results of the SVM, especially on the MNIST and COIL20 datasets, indicate that I-AE is able to embed the data in an arguably simpler, linearly separable manner. The very high classification rates in COIL20 are probably due to the size and structure of this dataset. Nevertheless with SVM, already in 16 dimensions I-AE provides an accuracy of $95\%$, with $5\%$ margin from 2nd place.

| Dataset | d | IAE | AE | CAE | DAE | RAE-GP | | Dataset | d | IAE | AE | CAE | DAE | RAE-GP |
|---------|-----|--------|--------|--------|--------|--------|--|---------|-----|--------|--------|--------|--------|--------|
| MNIST | 16 | **0.9138** | 0.9044 | 0.9045 | 0.9039 | 0.9016 | | MNIST | 16 | **0.9791** | 0.9784 | 0.9759 | 0.9783 | 0.9778 |
| | 64 | **0.8905** | 0.8364 | 0.8869 | 0.8373 | 0.8796 | | | 64 | 0.9736 | 0.9756 | 0.9761 | 0.9768 | 0.9710 |
| | 256 | **0.9585** | 0.9140 | 0.9308 | 0.9226 | 0.9347 | | | 256 | 0.9761 | 0.9737 | 0.9719 | 0.9734 | 0.9613 |
| FMNIST | 16 | **0.7910** | 0.7827 | 0.7865 | 0.7843 | 0.7831 | | FMNIST | 16 | 0.8790 | 0.8775 | 0.8773 | 0.8778 | 0.8763 |
| | 64 | 0.8343 | 0.8056 | 0.8350 | 0.7925 | 0.8339 | | | 64 | 0.8845 | 0.8867 | 0.8855 | 0.8854 | 0.8873 |
| | 256 | **0.8721** | 0.8341 | 0.8622 | 0.8374 | 0.8688 | | | 256 | 0.8778 | 0.8796 | 0.8763 | 0.8783 | 0.8712 |
| COIL20 | 16 | **0.9500** | 0.8944 | 0.8917 | 0.9000 | 0.8833 | | COIL20 | 16 | 0.9917 | 0.9861 | 0.9889 | 0.9861 | 0.9889 |
| | 64 | **1.0000** | 0.9944 | 0.9972 | 0.9917 | 0.9833 | | | 64 | 0.9917 | 0.9778 | 0.9722 | 0.9778 | 0.9722 |
| | 256 | **1.0000** | 0.9972 | 1.0000 | 0.9972 | 1.0000 | | | 256 | 0.9889 | 0.9750 | 0.9806 | 0.9722 | 0.9444 |

SVM                                         $K$-NN

Table 2: Downstream classification experiment. Both tables indicate accuracy in $[0, 1]$. Left: results with a linear SVM classifier; and right: results of a $K$-NN classifier with $K$=5. The top performance scores are highlighted with colors: **first**, second and third.

### 4.4 HYPER-PARAMETERS SENSITIVITY

To evaluate the affect of $\lambda_{\text{iso}}$ on the output we compared the visualizations and optimized loss values of MNIST and FMNIST, trained with same CNN architecture as in Section 4.2 with $\lambda_{\text{iso}} \in \{0, 0.01, 0.025, 0.05, 0.075, 0.1, 0.25, 0.5, 0.75, 0.1\}$. Figure 6 shows the different visualization results as well as $L_{\text{rec}}, L_{\text{iso}}, L_{\text{piso}}$ as a function of $\lambda_{\text{iso}}$. As can be seen in both datasets the visualizations and losses are stable for $\lambda_{\text{iso}}$ values between $0.01$ and $0.5$, where a significant change to the embedding is noticeable at $0.75$. The trends in the loss values are also rather stable; $L_{\text{iso}}$ and $L_{\text{piso}}$ start very high in the regular AE, i.e., $\lambda_{\text{iso}} = 0$, and quickly stabilize. As for $L_{\text{rec}}$ on FMNIST we see a stable increase while in MNIST it also starts with a steady increase until it reaches $0.75$ and then it starts to be rockier, which is also noticeable in the visualizations.

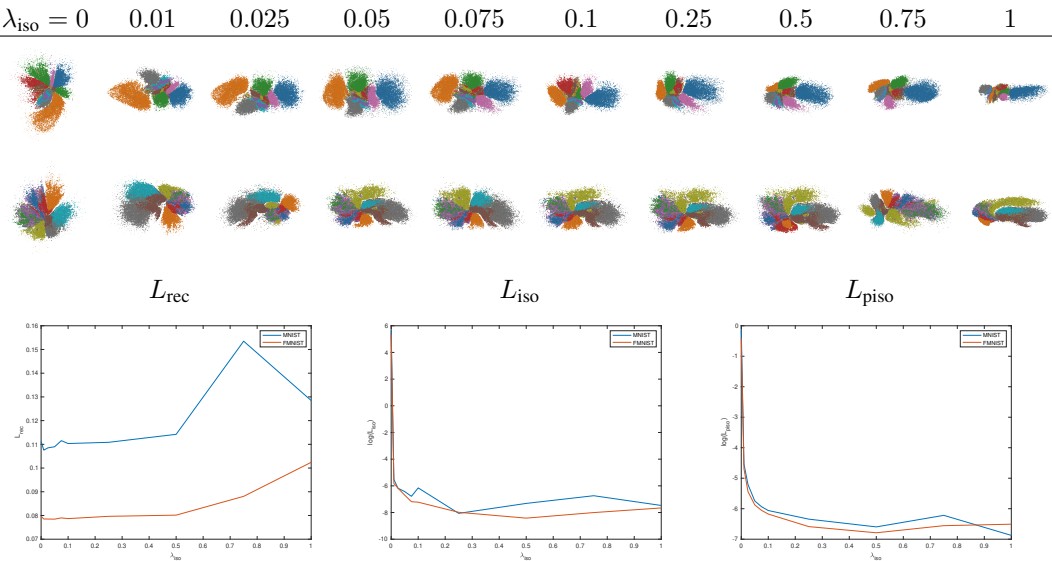

Figure 6: Sensitivity to hyper-parameters. Top: visualizations of MNIST (1st row) and FMNIST (2nd row) datasets trained with different $\lambda_{\mathrm{iso}}$ values. Bottom: plots of the final train losses as a function of $\lambda_{\mathrm{iso}}$; left to right: $L_{\mathrm{rec}}$ (linear scale), $L_{\mathrm{iso}}$ (log scale), and $L_{\mathrm{piso}}$ (log scale).

## 5 CONCLUSIONS

We have introduced I-AE, a regularizer for autoencoders that promotes isometry of the decoder and pseudo-inverse of the encoder. Our goal was two-fold: (i) producing a favorable low dimensional manifold approximation to high dimensional data, isometrically parameterized for preserving, as much as possible, its geometric properties; and (ii) avoiding complex isometric solutions based on the notion of psuedo-inverse. Our regularizers are simple to implement and can be easily incorporated into existing autoencoders architectures. We have tested I-AE on common manifold learning tasks, demonstrating the usefulness of isometric autoencoders.

An interesting future work venue is to consider task (ii) from section 1, namely incorporating I-AE losses in a probabilistic model and examine the potential benefits of the isometry prior for generative models. One motivation is the fact that isometries push probability distributions by a simple change of coordinates, $P(\boldsymbol{z}) = P(f(\boldsymbol{z}))$.

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

# A APPENDIX

## A.1 IMPLEMENTATION DETAILS

All experiments were conducted on a Tesla V100 Nvidia GPU using PYTORCH framework Paszke et al. (2017).

### A.1.1 NOTATIONS

Table 3 describes the notation for the different network layers.

| Notation | Description |
|---|---|
| LIN $n$ | Linear layer. $n$ denotes the output dimension. |
| FC $n$ | FullyConnected layer with SoftPlus ($\beta = 100$) non linear activation. $n$ denotes the output dimension. |
| FC_B $n$ | Block consisting of Lin $n$, followed by a batch normalization layer and SoftPlus ($\beta = 100$) non linear activation. |
| CONV $c, k, s, p$ | Convolutional layer with kernel of size $k \times k$, $c$ output channels, $s$ stride, and $p$ padding. |
| CONV_B $c, k, s, p$ | Block consisting of CONV $c, k, s, p$, followed by a batch normalization layer and SoftPlus($\beta = 100$) non linear activation. |
| CONVT $c, k, s, p$ | Convolutional transpose layer with kernel of size $k \times k$, $c$ output channels, $s$ stride, and $p$ padding. |
| CONVT_B $c, k, s, p$ | Block consisting of CONVT $c, k, s, p$, followed by a batch normalization layer and SoftPlus($\beta = 100$) non linear activation. |

Table 3: Layers notation.

### A.1.2 EVALUATION

**Architecture.** We used an autoencoder consisted of 5 FC 256 layers followed by a LIN 2 layer for the encoder; similarly, 5 FC 256 layers followed by a LIN 3 layer were used for the decoder.

**Training details.** All methods were trained for a relatively long period of 100K epochs. Training was done with the ADAM optimizer Kingma & Ba (2014), setting a fixed learning rate of 0.001 and a full batch. I-AE parameter was set to $\lambda_{\text{iso}} = 0.01$.

**Baselines.** The following regularizers were used as baselines: Contractive autoencoder (CAE) Rifai et al. (2011b); Contractive autoencoder with decoder weights tied to the encoder weights (TCAE) Rifai et al. (2011a); Gradient penalty on the decoder (RAE-GP) Ghosh et al. (2020); Denoising autoencoder with gaussian noise (DAE) Vincent et al. (2010). For both CAE, and TCAE the regularization term is $\|dg(\boldsymbol{x})\|^2$. For RAE-GP the regularization term is $\|df(\boldsymbol{z})\|^2$. For U-MAP McInnes et al. (2018), we set the number of neighbours to 30. For t-SNE Maaten & Hinton (2008), we set perplexity$= 50$.

### A.1.3 DATA VISUALIZATION

**Architecture.** Table 4 lists the complete architecture details of this experiment. Both MNIST and FMNIST were trained with FC-NN and S-CNN, and COIL20 was trained with L-CNN.

**Training details.** Training was done using ADAM optimizer Kingma & Ba (2014). The rest of the training details are on table 5.

| FC-NN | | S-CNN | | L-CNN | |
|---|---|---|---|---|---|
| Encoder | Decoder | Encoder | Decoder | Encoder | Decoder |
| FC_B 128 | FC_B 1024 | Conv_B 32,4,2,1 | FC 256 | Conv_B 128,4,2,1 | ConvT_B 2048,4,1,0 |
| FC_B 256 | FC_B 512 | Conv_B 64,4,2,1 | ConvT_B 128,4,1,0 | Conv_B 256,4,2,1 | ConvT_B 1024,4,2,1 |
| FC_B 512 | FC_B 256 | Conv_B 128,4,2,1 | ConvT_B 64,4,2,1 | Conv_B 512,4,2,1 | ConvT_B 512,4,2,1 |
| FC_B 1024 | FC_B 128 | Conv_B 256,4,2,0 | ConvT 32,4,2,1 | Conv_B 1024,4,2,1 | ConvT_B 256,4,2,1 |
| Lin 2 | Lin 784 | Lin 2 | ConvT 1,4,2,3 | Conv_B 2048,4,2,1 | ConvT_B 128,4,2,1 |
| | | | | Conv_B 4096,4,2,1 | ConvT 1,4,2,1 |
| | | | | Conv 2,2,2,1 | |

Table 4: High dimensional visualization experiment architectures.

| | MNIST | | FMNIST | | COIL20 |
|---|---|---|---|---|---|
| Architecture | FC-NN | S-NN | FC-NN | S-CNN | L-CNN |
| Batch Size | 128 | 512 | 128 | 512 | 144 |
| $\lambda_{\text{iso}}$ | 0.1 | 0.075 | 0.01 | 0.075 | 0.1 |
| Epochs | 1000 | 500 | 1000 | 500 | 1000 |

Table 5: High dimensional visualization training details.

**Baselines.** The following regularizers were used as baselines: Contractive autoencoder (CAE) Rifai et al. (2011b); Gradient penalty on the decoder (RAE-GP) Ghosh et al. (2020); Denoising autoencoder with gaussian noise (DAE) Vincent et al. (2010). For CAE the regularization term is $\|dg(\boldsymbol{x})\|^2$. For RAE-GP the regularization term is $\|df(\boldsymbol{z})\|^2$. We used U-MAP McInnes et al. (2018) official implementation with random_state $= 42$, and Ulyanov (2016) multicore implementation for t-SNE Maaten & Hinton (2008) with default parameters.

## A.2 ADDITIONAL EXPERIMENTS

### A.2.1 GENERALIZATION IN HIGH DIMENSIONAL SPACE

Next, we evaluate how well our suggested isometric prior induces manifolds that generalizes well to unseen data. We experimented with three different images datasets: MNIST (LeCun, 1998); CIFAR10 (Krizhevsky et al., 2009); and CelebA (Liu et al., 2015). We quantitatively estimate methods performance by measuring the $L_2$ distance and the *Fréchet Inception Distance* (FID) Heusel et al. (2017) on a held out test set. For each dataset, we used the official train-test splits.

For comparison versus baselines we have selected among relevant existing AE based methods the following: Vanilla AE (AE); autoencoder trained with weight decay (AEW); Contractive autoencoder (CAE); autoencoder with spectral weights normalization (RAE-SN); and autoencoder with $L_2$ regularization on decoder weights (RAE-SN). RAE-$L_2$ and RAE-SN were recently successfully applied to this data in (Ghosh et al., 2020), demonstrating state-of-the-art performance on this task. In addition, we compare versus the Wasserstein Auto-Encoder (WAE) Tolstikhin et al. (2018), chosen as state-of-the-art among generative autoencoders.

For evaluation fairness, all methods were trained using the same training hyper-parameters: network architecture, optimizer settings, batch size, number of epochs for training and learning rate scheduling. See the appendix for specific hyper-parameters values. In addition, we generated a validation set out of the training set using 10k samples for the MNIST and CIFAR-10 experiment, whereas for the CelebA experiment we used the official validation set. For each training epoch, we evaluated the reconstruction $L_2$ loss on the validation set and chose the final network weights to be the one that achieves the minimum reconstruction. We experimented with two variants of I-AE regularizers: $L_{\text{piso}}$ and $L_{\text{piso}} + L_{\text{iso}}$. Table 7 logs the results. Note that I-AE produced competitive results with the current SOTA on this task.

**Architecture.** For all methods, we used an autoencoder with Convolutional and Convolutional transpose layers. Table 6 lists the complete details.

**Training details.** Training was done with the ADAM optimizer Kingma & Ba (2014), setting a learning rate of 0.0005 and batch size 100. I-AE parameter was set to $\lambda_{\text{iso}} = 0.1$.

| MNIST | | CIFAR-10 | | CelebA | |
|---|---|---|---|---|---|
| Encoder | Decoder | Encoder | Decoder | Encoder | Decoder |
| Conv_B 128, 4, 2, 1 | FC 16384 | Conv_B 128, 4, 2, 1 | FC 16384 | Conv_B 128, 5, 2, 1 | FC 65536 |
| Conv_B 256, 4, 2, 1 | ConvT_B 512, 4, 2, 1 | Conv_B 256, 4, 2, 1 | ConvT_B 512, 4, 2, 1 | Conv_B 256, 5, 2, 1 | ConvT_B 512, 4, 2, 1 |
| Conv_B 512, 4, 2, 1 | ConvT_B 256, 4, 2, 1 | Conv_B 512, 4, 2, 1 | ConvT_B 256, 4, 2, 1 | Conv_B 512, 5, 2, 1 | ConvT_B 256, 4, 2, 1 |
| Conv_B 1024, 4, 2, 1 | ConvT_B 128, 4, 2, 1 | Conv_B 1024, 4, 2, 1 | ConvT_B 128, 4, 2, 1 | Conv_B 1024, 5, 2, 1 | ConvT_B 128, 4, 2, 1 |
| Lin 16 | ConvT 1, 1, 0, 0 | Lin 128 | ConvT 3, 1, 0, 0 | Lin 128 | ConvT 3, 1, 0, 0 |

Table 6: High dimensional generalization experiment architectures.

| Dataset | Distance | Methods | | | | | | | |
|---|---|---|---|---|---|---|---|---|---|
| | | $L_{\text{piso}}$ | $L_{\text{piso}} + L_{\text{iso}}$ | AE | AEW | CAE | RAE-SN | RAE-$L_2$ | WAE |
| MNIST | $L_2$ | **0.96** | 0.99 | 1.14 | 1.0 | 1.15 | 1.35 | 1.14 | 1.64 |
| | FID | 6.09 | 7.94 | **4.95** | 5.59 | 6.46 | 10.72 | 11.41 | 6.99 |
| CIFAR-10 | $L_2$ | 20.19 | 21.05 | **20.16** | 20.33 | 20.23 | 21.02 | 20.2 | 21.08 |
| | FID | 70.14 | **56.04** | 74.79 | 68.71 | 71.71 | 70.79 | 71.05 | 74.2 |
| CelebA | $L_2$ | 20.38 | 19.93 | 20.51 | **19.74** | 20.46 | 20.78 | 20.58 | 20.88 |
| | FID | **34.68** | 40.73 | 40.53 | 40.00 | 39.52 | 40.45 | 38.86 | 38.98 |

Table 7: Manifold approximation quality on test images. We log the $L_2$ and FID distances (lower is better) from reconstructed images to the input images. The $L_2$ numbers are reported $*10^3$. The top performance scores are highlighted as: **First**, Second.

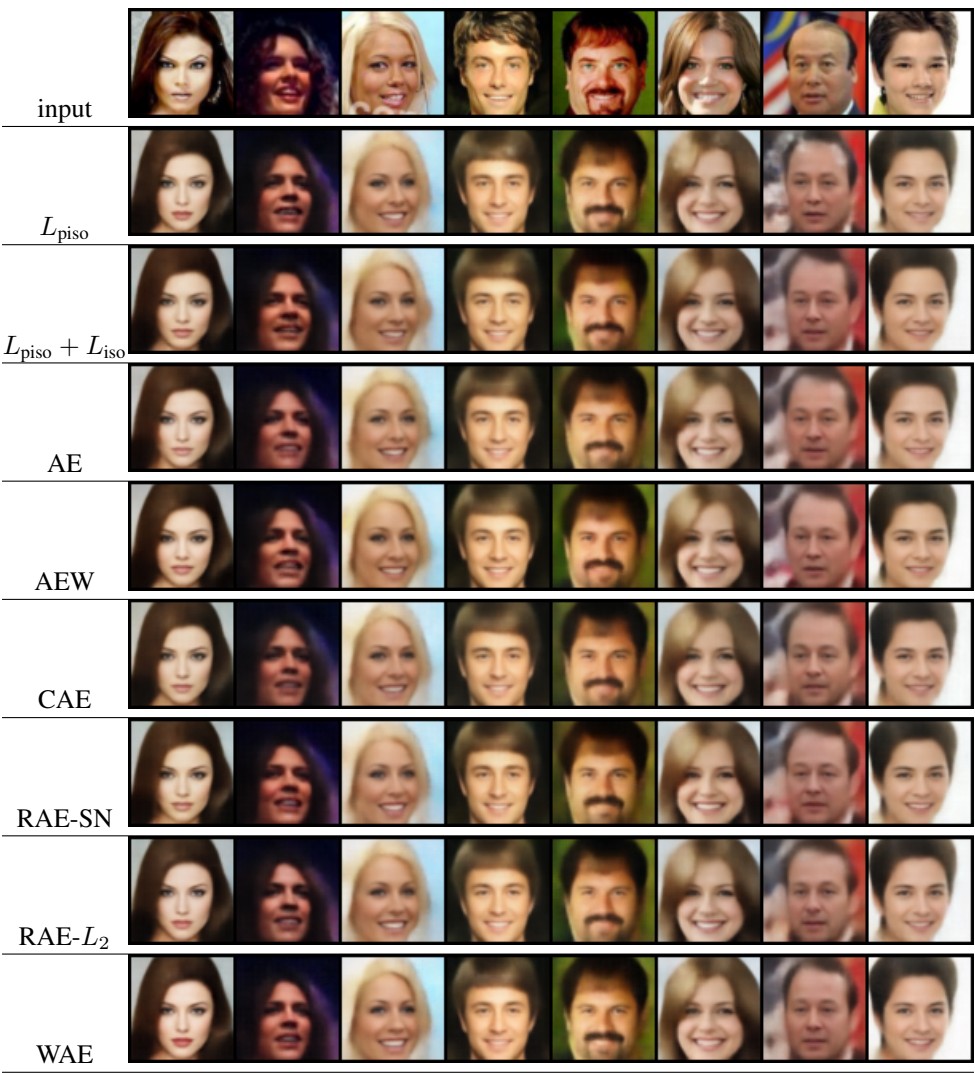

Figure 7: CelebA reconstructions.

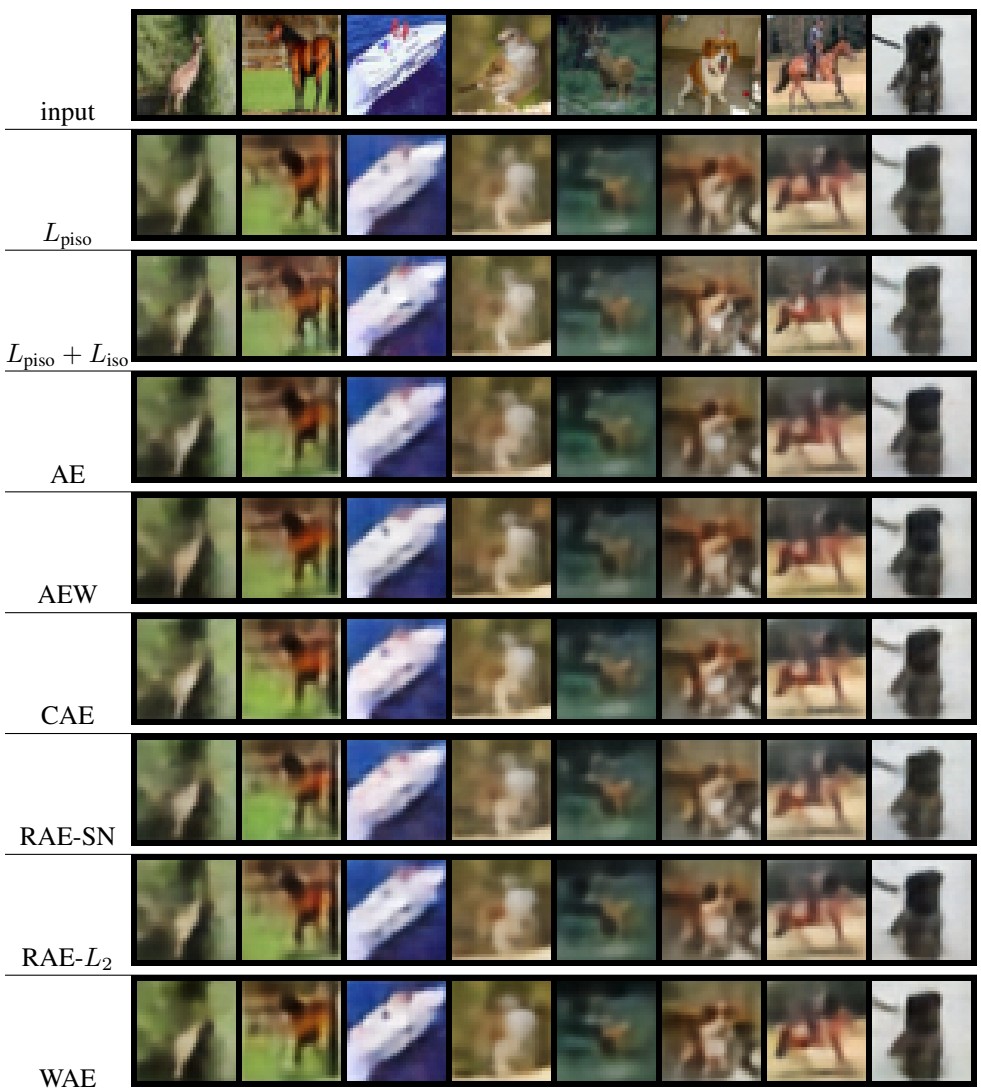

Figure 8: CIFAR-10 reconstructions.

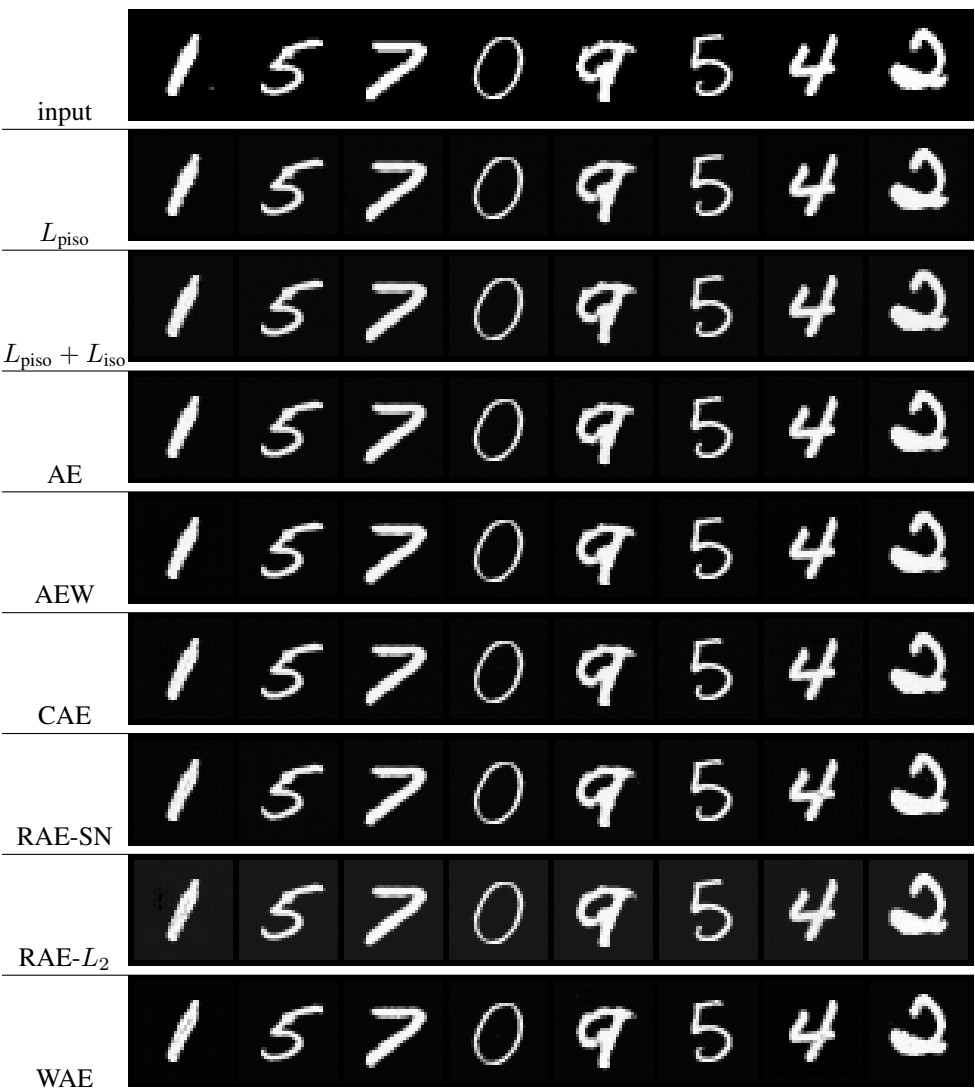

Figure 9: MNIST reconstructions.

