# OpenReview forum: "Isometric Autoencoders"
_ICLR.cc/2021/Conference — Reject_

### Official Review · AnonReviewer3 · 2020-10-28
**The theories are nice but the experiment doesn't support the theory enough.**

**Rating:** 6
**Confidence:** 4

**Review:**

Strength:
1. This paper provided a novel method to train a local isometric autoencoder, which can preserve the local Euclidean distances well between the original space and the latent space.
2. The theories are well presented and explained pretty well. Also, Isometry is a very important property in several cases including manifold-learning, etc.
3. Apart from the typos and several tiny errors, the overall writing is sound and smooth.

Weakness:
1. I have to say that the argument of global isometry is too ambitious. In the theory and method part, this method only guarantees the local isometry. The authors do mention that "a local isometry which is also a diffeomorphism is a global isometry" on page 3 bottom paragraph. However, there's no discussion about the "diffeomorphism" in the following sections.
2. Also, the first experiment (3D $\rightarrow$ 2D) only supports the local isometry. Due to the fact that the distance is computed only based on the triangular meshes between edges. Is there any possibility that the author can provide one more toy example for the global isometry when the data lie on some manifold shape? This would strongly support the global argument.
3. From my understanding, the distance in the original space is the Euclidean distance without considering the local geometry of the data. Can the author provide some comments on this? The distance in the original space should be the geodesic distance when arguing the Isometry.
4. The experiment of the data visualization is somewhat weak. The benefit of the Isometry Autoencoder is not well addressed. The t-SNE is well used for visualization with almost nothing wrong. The only benefit comes when arguing the "even" sampling. Can the author provide some comments on this about why we need the "even" in the visualization?

Some other minor comments:
1. There's a typo in the last line of the first page. The encoder should be $R^D\rightarrow R^d$ with $d<D$. Similar to the inverse $R^d\rightarrow R^D$.
2. The differential of the decoder should be the Jacobian matrix right? This would be more clear than just mentioning differential.
3. Table 1 should be Figure 1. Also, can the author provide more details about this figure? Is this figure for illustration only or this result is actually trained and plotted? The form "evenly" is a strong word that needs more explanation of the definition.
4. The order of Figure 3 and Figure 2 is messed up.

---

> ### Author Response · Authors · 2020-11-20
> **Authors response to reviewer 3**
>
> **Q: I have to say that the argument of global isometry is too ambitious. In the theory and method part, this method only guarantees the local isometry. The authors do mention that "a local isometry which is also a diffeomorphism is a global isometry" on page 3 bottom paragraph. However, there's no discussion about the "diffeomorphism" in the following sections.**
>
> **A:** The diffeomorphism of the decoder is encouraged by the reconstruction term, as mentioned at the bottom of page 4 below Lemma 2. Therefore as the decoder and the encoder are smooth by construction (we use Softplus instead of ReLU) we are encouraging our decoder to be a diffeomorphism.
>
> **Q: Is there any possibility that the author can provide one more toy example for the global isometry when the data lie on some manifold shape? This would strongly support the global argument.**
>
> **A:** We would like to point out that figure 3, and table 1 already provide such an example. In figure 3, the S surface and the swiss-roll surfaces are both isometric to the euclidean plane. Isometries in particular preserve densities (unit det jacobian) and since the surfaces are sampled uniformly (left column), global isometry can be inspected by observing the (almost) perfect planar rectangle with uniformly sampled points, as shown in the I-AE column. This qualitatively demonstrates the (global) isometry. Table 1, quantitatively demonstrates low isometric distortion that, together with the bijectivity in these examples, implies the encoder is close to a perfect (global) isometry.
>
> **Q: From my understanding, the distance in the original space is the Euclidean distance without considering the local geometry of the data. Can the author provide some comments on this? The distance in the original space should be the geodesic distance when arguing the Isometry.**
>
> **A:** The distance in the original space is **not** the euclidean distance. The decoder $f$ is encouraged to have an orthogonal differential (equation 6). This local condition, if satisfied everywhere, guarantees isometry, that is geodesic distances on the manifold are represented by straight lines in the latent space.
>
> **Q: The experiment of the data visualization is somewhat weak. The benefit of the Isometry Autoencoder is not well addressed. The t-SNE is well used for visualization with almost nothing wrong. The only benefit comes when arguing the "even" sampling. Can the author provide some comments on this about why we need the "even" in the visualization?**
>
> **A:** The “even” structure is useful when the data is uniformly sampled from the data manifold. For example, in the COIL20 dataset, we expect to get equidistant rings, i.e., with even spacings. Notice that t-SNE is limited only to embed data points and does not generalize to unseen data, as opposed to autoencoders. Lastly, to further highlight the benefit in the isometric AE beyond visualization, we provide an additional experiment in the revision, where we use the (unsupervised) embedded vectors to train a simple classifier (KNN or linear SVM), see section 4.3 in the revised paper. As can be seen in that experiment, I-AE embeddings result in higher accuracies of the simple classifiers, over a wide range of parameters (such as latent dimension). The fact that I-AE outperforms other AEs showcase its ability to better preserve and model the structure of the high dimensional data manifold.

---

### Official Review · AnonReviewer1 · 2020-10-28

**Rating:** 4
**Confidence:** 3

**Review:**

The authors propose a new version of the regularized autoencoder where they explicitly regularizes its decoder to be locally isometric and its encoder to be the decoder's pseudo inverse. Through a series of experiments and visualization, the IAE exhibits better manifold structure.

Regarding the motivation and the math, I like the idea of isometric regularizer preserving the geometric properties in the learned manifold. The illustration in figure 1 does clearly point out the advantages of IAE over the contractive autoencoder. The math formulation primarily sticks with a linear version of the autoencoder. It would be great to get some insights for a non-linear counterpart.

Regarding the experiments, indeed the authors successfully show the IAE converges its decoder to be an isometry and the proposed regularizer promotes more favoured manifold. However, the experiments mainly rely on visualization but fail to give some numeric results. For instance, can IAE be useful for semi-supervised learning (Like VAEs)? How can we practically make use of the isometry property in applications other than data visualization?

---

> ### Author Response · Authors · 2020-11-20
> **Authors response to reviewer 1**
>
> **Q: The experiments mainly rely on visualization but fail to give some numeric results. For instance, can IAE be useful for semi-supervised learning (Like VAEs)? How can we practically make use of the isometry property in applications other than data visualization?**
>
> **A:** Thank you for this comment. We provide an additional experiment in the revision, where we use the (unsupervised) embedded vectors to train a simple classifier (KNN or linear SVM), see section 4.3 in the revised paper. As can be seen in that experiment, I-AE embeddings result in higher accuracies of the simple classifiers, over a wide range of parameters (such as latent dimension). The fact that I-AE outperforms other AEs showcase its ability to better preserve and model the structure of the high dimensional data manifold.

---

### Official Review · AnonReviewer4 · 2020-10-29
**Novel auto-encoder based method for manifold learning**

**Rating:** 6
**Confidence:** 2

**Review:**

The paper suggests a novel auto-encoder based method for manifold learning, by encouraging the decoder to be an isometry and the encoder to locally be a pseudo-inverse of the decoder.  It is noted that for a linear architecture, this gives PCA, therefore, this can be seen as a nonlinear PCA approach.

In theorem 1, the authors claim that for the encoder-decoder solution to have the desired properties, certain equalities have to be satisfied by the local differential matrices of the encoder and decoder.  This gives rise to a loss function that is combined of 3 parts:  A reconstruction loss (as usual with autoencoders) plus a combination of a loss penalizing non isometric decoders, plus a loss penalizing an encoder that is not a pseudo-inverse of the decoder.  This loss function is claimed to be the main technical novelty of the paper.

In the experimental part, the authors compare the merits of this approach on synthetically generated low dimensional manifolds in high dimensional ambient spaces, against other standard manifold learning algorithms, and show that the paper's method outperforms other method using a measure of distortion of triangle edges on a grid.  They also experiment with "real data" (e.g. MNIST), show the merits of the proposed algorithm when visualizing the 2 dimensional bottleneck of the autoencoder.  The comparison here is against other algorithms for high dimensional data visualization.

The overall idea and theory seem interesting.  The experiments are a bit disappointing.  For the synthetic data, I am not sure I understand why they did not chose something of high dimension?  Maybe I am missing something, but would it be impossible to generate, say, a 50 dimensional manifold in 100 dimensions?  Maybe the triangulation part will be challenging, but that is not the only way to compare between the various algorithms.  As for the real data section (e.g. MNIST), I am not sure I see why you compare your algorithm against algorithms that are intended for 2-d visualization (e.g. t-SNE).  Your algorithm does manifold learning.  Why not,for instance, take all the images corresponding to some fixed digit (e.g. "3"), which is presumably close to a low (but definitely more than 2....)  dimensional manifold, and see how well your manifold learning algorithm reconstructs them?

The editorial level of the paper is not very high, due to grammatical English mistakes.  Here are examples (the list is not complete):
p. 1 "Autoencoder (AE) can also be seen" => "Autoencoders can also be seen" or "An  autoencoder can also be seen..."

"AE is trying to reconstruct X..."  - The present progressive tense is not suitable here.  Maybe "AE's try to reconstruct"?  Or "AE's are designed to reconstruct..." or "An AE reconstructs..."

p. 2
Manifold learning generalizeS


p. 4
"As-usual " => As usual

p. 5
"Does our suggested Loss... drives" -> "drive"

p. 6
Why is "Denoising" capitalized?

"In addition, we compared versus..." => "...compared against..."

---

> ### Author Response · Authors · 2020-11-20
> **Authors response to reviewer 4**
>
> **Q: For the synthetic data, I am not sure I understand why they did not chose something of high dimension? Maybe I am missing something, but would it be impossible to generate, say, a 50 dimensional manifold in 100 dimensions? Maybe the triangulation part will be challenging, but that is not the only way to compare between the various algorithms.**
>
> **A:** We experiment with standard manifold examples commonly used in existing literature. The surfaces chosen already pose a challenge to previous methods while allowing quantitative and qualitative evaluation. We believe triangulating a high dimensional manifold would be exponential in the dimension and therefore very challenging.
>
> **Q: Your algorithm does manifold learning. Why not, for instance, take all the images corresponding to some fixed digit (e.g. "3"), which is presumably close to a low (but definitely more than 2....) dimensional manifold, and see how well your manifold learning algorithm reconstructs them?**
>
> **A:** Thank you for this comment. To quantitatively evaluate how well our algorithm learns manifolds of higher dimension we provided an additional experiment in the revision, where we use the embedded vectors to train a simple classifier (KNN or linear SVM), see section 4.3 in the revised paper. The fact that I-AE outperforms other AEs showcase its ability to better preserve and model the structure of the high dimensional data manifold.

---

### Official Review · AnonReviewer2 · 2020-11-04
**Official Blind Review #2**

**Rating:** 7
**Confidence:** 4

**Review:**

Update: I appreciate the authors addressing my concerns. I have increased my score accordingly.

Original Review:

This paper describes a new type of regularization for the parameters of an autoencoder - one that forces the decoder to be an isometry. The authors present conditions that need to be satisfied by the encoder and decoder parameters, and show empirically that the regularization terms that they propose ensure that the resulting autoencoder has an isometric decoder. The paper is well written and easy to follow.

While the authors assert that forcing the decoder to be an isometry is desirable since isometries preserve distances and angles, it is not clear why that is a desirable property while modeling data on a manifold. Distances between points on a data manifold are not usually measured through L2 distances in a latent dimension, and it is not clear why one should require that L2 distances in the high dimensional space are the same as distances in the latent space. The numerical results on reconstruction error that the authors present in the appendix do not indicate any reason to prefer isometric AEs over other baselines that are considered. In case there is a setting where isometric AEs can be shown to model the data manifold better than regular AEs, that is not highlighted in the current draft.

The authors claim that isometric autoencoders would "evenly sample the manifold" which is a little confusing, since the sampling of the data manifold is separate from the technique used to model the data (regular AEs vs isometric AEs).

The experimental results also do not indicate how the embeddings learned using the proposed method perform on downstream classification tasks, for instance. This comparison would be useful to have to compare the usefulness of the embeddings.

A few minor points of confusion:
1) the notation f^{-1} is a little misleading since the encoder is not necessarily an invertible function from R^d to R^D. If the encoder mapping is restricted to the range of f then this notation is more appropriate.
2) The projection operator that is used to define the pseudoinverse of the encoder is not necessarily a function, since there could possibly be many points on the manifold that correspond to the same L2 distance from the point being projected. Are there further assumptions on the structure of the data manifold that prevent this from being the case?
3) Estimating the L_iso term seems to require a distribution over the latent space R^d, that the authors say is computed using a fit of the latent codes g(x), x \in \cal X. Are the latent codes computed using the current estimate of the encoder? If so is there some sort of alternating minimization happening, which holds the current estimate of the encoder fixed while computing the isometric regularization? If not, how are the latent codes computed?

---

> ### Author Response · Authors · 2020-11-20
> **Authors response to reviewer 2**
>
> **Q: The experimental results also do not indicate how the embeddings learned using the proposed method perform on downstream classification tasks, for instance. This comparison would be useful to have to compare the usefulness of the embeddings.**
>
> **A:** Thank you for this comment. We have added an additional experiment in the revised paper (section 4.3), that evaluates our embeddings for downstream classification tasks. As you can see in the results, we outperform other autoencoder methods.
>
> **Q: It is not clear why one should require that L2 distances in the high dimensional space are the same as distances in the latent space.**
>
> **A:** Isometry does not mean L2 distances in the high dimensional space are preserved in the latent space; rather, **geodesic distances** over the manifold are preserved. The geodesic distances indeed locally coincide with the L2 metric in the ambient Euclidean space, however, the geodesic distance for distant points measures the shortest path restricted to the manifold.
>
> **Q: The numerical results on reconstruction error that the authors present in the appendix do not indicate any reason to prefer isometric AEs over other baselines that are considered. In case there is a setting where isometric AEs can be shown to model the data manifold better than regular AEs, that is not highlighted in the current draft.**
>
> **A:** We refer the reviewer again to section 4.3 in the revised paper for another quantitative justification showing isometric AEs better model manifold data compared to other AEs.
>
> **Q: The authors claim that isometric autoencoders would "evenly sample the manifold" which is a little confusing, since the sampling of the data manifold is separate from the technique used to model the data (regular AEs vs isometric AEs).**
>
> **A:** We meant that isometric autoencoders evenly sample the manifold in the sense they do not shrink or expand the space, locally they behave as orthogonal linear transformations. We added a clarification in the paper.
>
> **Q: The projection operator that is used to define the pseudoinverse of the encoder is not necessarily a function, since there could possibly be many points on the manifold that correspond to the same L2 distance from the point being projected. Are there further assumptions on the structure of the data manifold that prevent this from being the case?**
>
> **A:** For closed manifolds there is always the closest point, although indeed not necessarily unique. So technically the projection can be made a function (i.e., choose one closest point), although not continuous everywhere. In any case, for points close enough to a smooth manifold uniqueness holds. We added a clarification in the revised paper (see just before Definition 1).
>
> **Q: Estimating the L_iso term seems to require a distribution over the latent space R^d, that the authors say is computed using a fit of the latent codes g(x), x \in \cal X. Are the latent codes computed using the current estimate of the encoder?**
>
> **A:** During training, for each batch we compute the mean and standard deviation of the (current) encoded batch, and use it to define a multivariate gaussian, from which we sample.

---

### Decision · Program_Chairs · 2021-01-07
**Final Decision**

**Decision:**

Reject

**Comment:**

The paper introduces a new formulation for learning low-dimensional manifold representations via autoencoder mappings that are (locally) isometric by design. The key technical ingredient is the use of a particular (theoretically motivated) weight-tied architecture coupled with isometry-promoting loss terms that can be approximated via Monte Carlo sampling. Representative results on simple manifold learning experiments are shown in support of the proposed formulation.

The paper was generally well-received; all reviewers appreciated the theoretical elements as well as the presentation of the ideas.

However, there were a few criticisms. First, the fact that the approach requires Monte Carlo sampling in very high dimensions automatically limits its scope. Second, the experiments seemed somewhat limited to simple (by ICLR standards) datasets. Third and most crucially, the approach lacks a compelling-enough use case. It is not entirely clear what local isometry enables, beyond nice qualitative visualizations (and moreover, what the isometric autoencoder provides over other isometry-preserving manifold learning procedures such as ISOMAP). Some rudimentary results are shown on k-NN classification and linear SVMs, but the gains seem to be in the margins.

The authors are encouraged to consider the above concerns (and in particular, identifying a unique use case for isometric autoencoders) while preparing a future revision.